# EVOLUTIONARY-NEURAL HYBRID AGENTS FOR ARCHITECTURE SEARCH

## ABSTRACT

Neural Architecture Search has recently shown potential to automate the design of Neural Networks. The use of Neural Network agents trained with Reinforcement Learning can offer the possibility to learn complex patterns, as well as the ability to explore a vast and compositional search space. On the other hand, evolutionary algorithms offer the greediness and sample efficiency needed for such an application, as each sample requires a considerable amount of resources. We propose a class of Evolutionary-Neural hybrid agents (Evo-NAS), that retain the best qualities of the two approaches. We show that the Evo-NAS agent can outperform both Neural and Evolutionary agents, both on a synthetic task, and on architecture search for a suite of text classification datasets.

## 1 INTRODUCTION

Deep Neural Networks (DNN) have yielded success in many supervised learning problems. However, the design of state-of-the-art deep learning algorithms requires many decisions, normally involving human time and expertise. As an alternative, Auto-ML approaches aim to automate manual design with meta-learning agents. Many different approaches have been proposed for architecture optimization, including random search, evolutionary algorithms, Bayesian optimization and DNN trained with Reinforcement Learning.

Deep Reinforcement Learning (deep RL) is one of the most common approaches. It samples architectures from a distribution, which is modeled by a deep neural network. The parameters of the controller model are trained using RL to maximize the performance of the downstream learning algorithm. Architecture search based on deep RL has shown many recent successes in automatic design of state-of-the-art RNN cells (Zoph & Le, 2017), convolutional blocks (Zoph et al., 2017), activation functions (Prajit Ramachandran, 2018), optimizers (Bello et al., 2017; Wichrowska et al., 2017) and data augmentation strategies (Cubuk et al., 2018).

Recently, Real et al. (2018) has shown that evolutionary approaches, with appropriate regularization, can match or surpass the RL-DNN meta learner on architecture search task where sample efficiency is critical. Indeed, those methods can efficiently exploit a single observation of a good trial, when learning based methods require more observations. Classical evolutionary methods have the disadvantage of relying on heuristics or random sampling when choosing mutations to be applied to the parent trial. Unlike approaches based on a Neural Network (NN) controller, they are unable to learn patterns to drive the search.

The scientific contribution of this paper is to introduce a class of Evolutionary-Neural hybrid agents (Evo-NAS). Concretely, we propose an evolutionary agent, where the mutations are guided by a NN trained with RL. This combines both the sample efficiency of Evolutionary agents, and the ability to learn complex patterns, retaining the best of both worlds.

In Section 3 we give a brief description of state-of-the-art Neural and Evolutionary agents, and introduce by comparison the proposed Evo-NAS agent in Section 4. Then, in Section 5, we present and discuss the properties of the proposed Evolutionary-Neural agent, applying it to a synthetic task. Finally, we apply Evo-NAS to architecture search benchmarks, showing that it outperforms both RL-based and Evolution-based algorithms on architecture search for a variety of text classification datasets.

## 2 RELATED WORK

In recent years, progress has been made in automating the design process required to produce state-of-the-art neural networks. Recent methods have shown that learning-based approaches can achieve state-of-the-art results on ImageNet (Zoph & Le, 2017; Liu et al., 2017). These results have been subsequently scaled by transferring architectural building blocks between datasets (Zoph et al., 2017). Some works explicitly address resource efficiency (Zhong et al., 2018; Pham et al., 2018), which is crucial, as architecture search is known to require vast amount of resources (Zoph & Le, 2017).

Another important approach to architecture search is neuro-evolution (Floreano et al., 2008; Stanley et al., 2009; Real et al., 2017; Conti et al., 2017). Recent work has highlighted the importance of regularizing the evolutionary process, showing that evolution can match or outperform a learning-based baseline (Real et al., 2018). Others have applied genetic algorithms to evolve the weights of the model (Such et al., 2017).

Other than deep RL and evolution, different approaches have been applied to architecture search: cascade-correlation (Fahlman & Lebiere, 1990), boosting (Cortes et al., 2016), deep-learning based tree search (Negrinho & Gordon, 2017), hill-climbing (Elsken et al., 2017) and random search (Bergstra & Bengio, 2012).

## 3 BASELINES

### 3.1 NEURAL ARCHITECTURE SEARCH

Neural Architecture Search (NAS) (Zoph & Le, 2017) uses a RNN controller to perform architectural choices that define a child model. These choices can also include hyperparameters such as learning rate. The resulting child model is then trained on the downstream task, and its accuracy on a validation set is computed. This accuracy serves as the reward for training the RNN controller using a policy gradient approach. In the following sections, we will refer to the standard NAS agent as the Neural agent.

### 3.2 REGULARIZED EVOLUTION ARCHITECTURE SEARCH

Regularized Evolution (Real et al., 2018) is a variant of the tournament selection method (Goldberg & Deb, 1991). A population of $P$ trained models is improved in iterations. At each iteration, a sample of $S$ models is selected at random. The best model of the sample is mutated to produce a child model with an altered architecture, which is trained and added to the population. The regularization consists of discarding the oldest model of the population instead of the one with the lowest reward. This avoids over-weighting of models that may have reached their high accuracy by chance during the noisy training process. In the following sections, we will refer to the standard Regularized Evolution agent as the Evolutionary agent.

## 4 EVOLUTIONARY-NEURAL ARCHITECTURE SEARCH

We propose a hybrid agent, which combines an Evolutionary approach with the use of a RL trained Neural controller. The population update and parent sampling is equivalent to Regularized Evolution.

The main difference is in how the mutations are generated. At mutation sampling time, the parent sequence is injected in a standard NAS-RNN controller. The Evolutionary-Neural controller has only one modification, that allows it either to reuse the parent's action or to sample a new one from the learned distributions over the action space. This decision to keep or re-sample an action is taken independently for any action in the sequence. If an action is re-sampled, it will be conditioned on all the prior actions as in the standard NAS-RNN controller. Concretely, the parent's action is reused with probability $p$, and a new action is sampled otherwise, where $p$ is a hyperparameter. This process is summarized in Figure 1.

At training time, the controller parameters are updated in the same way as in the standard NAS-RNN controller.

Figure 1: Overview of the way in which the Evo-NAS agent creates a child trial, given a parent trial. The colored blocks represent actions that the agent has to perform (e.g. architectural and hyperparameter choices). Each action is reused from the parent trial with probability $p$, or sampled from the distribution learned by the controller neural-network with probability $1 - p$.

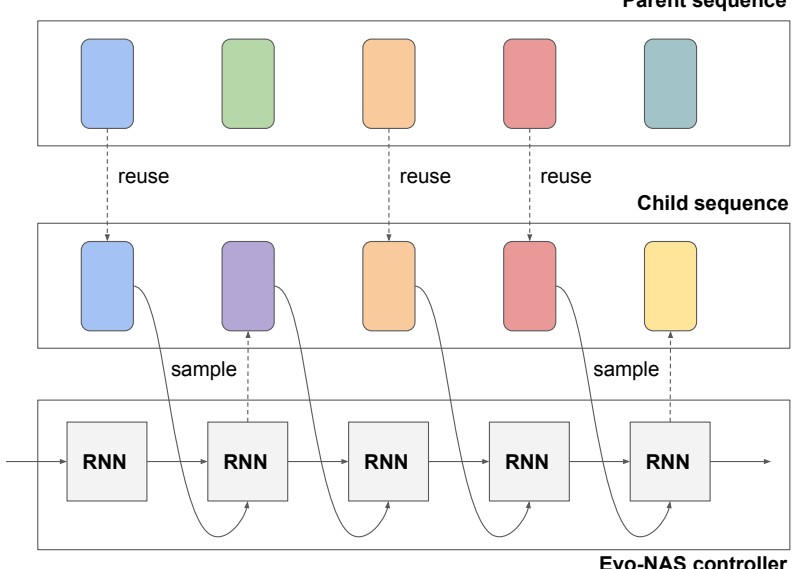

Note that, compared to Regularized Evolution, this design just affects the distributions from which the mutations are sampled. Regularized Evolution is unable to learn good mutation patterns, since the distributions from which the mutations are sampled are constantly uniform (random sampling). For Evo-NAS, these distributions are initialized as uniform, but with time are updated to promote the patterns of the good models.

Evo-NAS is designed to retain the sample-efficiency of Evolution, which is able to efficiently exploit good trials, while the underlining Neural Network is able to learn biases and complex patterns, as the standard Neural agent.

## 4.1 TRAINING

We consider two alternative training algorithms:

**Reinforce** (Williams, 1992). Reinforce is the standard on-policy policy-gradient training algorithm. It is often considered the default choice for applications where a NN needs to be trained to learn a task that has no supervision, but requires to explore a complex space to find the optimal solution, as is in the case of architecture search. This approach has the disadvantage of not being as sample efficient as its off-policy alternatives, which are able to reuse samples. Reinforce is the training algorithm chosen in the original NAS paper (Zoph & Le, 2017).

**Priority Queue Training** (PQT) (Abolafia et al., 2018). In PQT, samples are generated by sampling from the learned distributions as in Reinforce. However, at training time, the gradients are generated to directly maximize the log likelihood of the best samples produced so far. This training algorithm has the sample efficiency of RL off-policy training approaches, since the best models generate multiple updates. PQT has the simplicity of supervised learning, since the best models are directly promoted as if they constitute the supervised training set, with no need of reward scaling as in Reinforce, or sample probability scaling as in off-policy training.

In the experiment section we will compare these two training approaches and conclude that PQT leads to higher sample-efficiency and better results.

Table 1: Architecture search algorithms

| Algorithm | Learning | Direct exploitation |
|---|---|---|
| Random Search | No | No |
| Neural agent | Yes | No |
| Evolutionary agent | No | Yes |
| Evo-NAS agent | Yes | Yes |

## 5 EXPERIMENTS

To properly evaluate the advantages and disadvantages of different approaches, we propose to consider two characteristics:

- whether the agent is capable of learning patterns
- whether the agent is capable of efficiently exploiting good past experiences

The two characteristics above are independent, and for a fixed architecture search algorithm, both of them may or may not be present. In Table 1, we summarize the characteristics of the methods we aim to compare .

### 5.1 LEARN TO COUNT

To highlight the properties of the proposed Evo-NAS agent, we evaluated it on a synthetic task. In this task, the agent is asked to choose a sequence $\mathbf{a} = \langle a_1, a_2, \cdots, a_n \rangle$ of $n$ integer numbers. Each number is selected from the set $[1, n] \cap \mathbb{Z}$. Here $n$ is a hyperparameter that controls the difficulty of the task. Given a sequence, $\mathbf{a}$, its imbalance, $i(\mathbf{a})$, is defined as follows:

$$i(\mathbf{a}) = \sum_{k=0}^{n} (a_{k+1} - a_k)^2 \tag{1}$$

Where we set the initial and last terms $a_0 = 0$ and $a_{n+1} = n + 1$. The reward observed by the agent after choosing a sequence $\mathbf{a}$ is then:

$$r(\mathbf{a}) = \frac{n+1}{i(\mathbf{a})} \tag{2}$$

Notice that:

- in order to maximize reward, the agent must minimize imbalance.
- $r(\mathbf{a}) \in (0, 1]$.
- there is a single optimal sequence that achieves the reward of 1, namely $\mathbf{a} = \langle 1, 2, \cdots, n \rangle$.

The proposed toy task has multiple key properties:

- The size of the search space is $n^n$, which even for small $n$ is already too big for any exhaustive search algorithm to succeed.

- If $U$ is uniform distribution over $\{1, \cdots, n\}^n$ then: $\mathbb{E}_{a \sim U} [r(a)] = O\left(\frac{1}{n^2}\right)$.

  Moreover, Random Search performs very poorly, as shown by our experiments reported below. This allows to attribute the discovery of good sequences to properties of the algorithm, rather than to accidental discovery over time.

- The search space exhibits a sequential structure, with a notion of locally optimal decisions, making it a good task both for learning patterns, and for mutating past trials by local modifications.

Figure 2: Comparison between Reinforce and Priority-Queue-Training (PQT) on "Learn to count" synthetic task, for both Neural (Left) and Evo-NAS (Right) agents. The plot shows the best reward attained (Y-axis) after a given number of trials (X-axis). Each experiment was ran 20 times, and the shaded area represents 70% confidence interval.

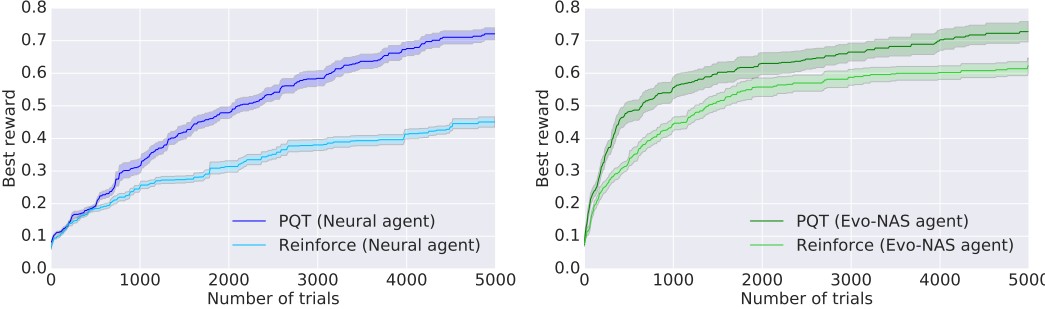

Figure 3: Comparison of different agents on "Learn to count" synthetic task. On the Y-axis, the plots show the moving average of the reward over 50 trials (Left) and the best reward attained so far (Right). Each experiment was ran 20 times, and the shaded area represents 70% confidence interval.

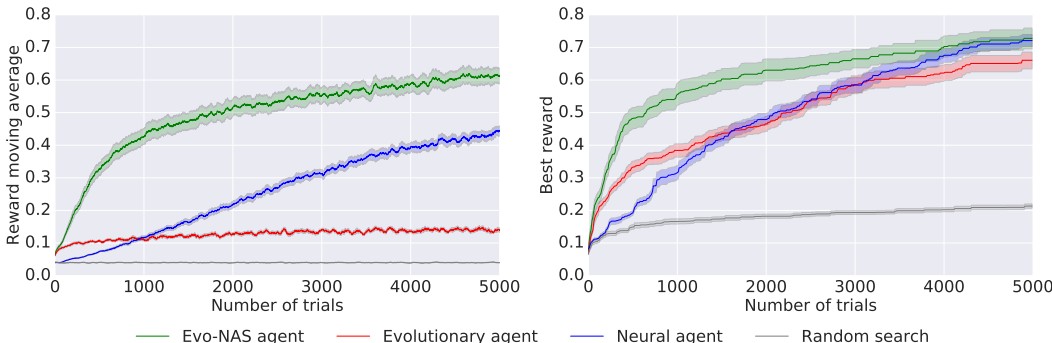

We perform a preliminary tuning of the hyperparameters of the agents to ensure a fair comparison. For the following experiments, we set the decision exploitation probability $p = 0.7$, the population size $P = 500$ and the sample size $S = 50$, for both the Evolutionary and the Evo-NAS agent. We set the learning rate to be 0.0005 for the Neural agent and 0.0001 for the Evo-NAS agent. We set the entropy penalty of 0.1 for the Neural agent and 0.2 for the Evo-NAS agent. PQT maximizes the log likelihood of the top 5% trials for the Neural agent and top 20% trials for the Evo-NAS agent.

We start by comparing Reinforce and PQT as alternative training algorithms for the Neural and Evo-NAS agents. The results of the experiments are shown in Figure 2. We found that PQT outperformed Reinforce for both the Neural and the Evo-NAS agent. For the Evo-NAS agent, the gain is especially pronounced at the beginning of the experiment. Thus, we conclude that PQT can provide a stronger training signal than Reinforce. We will use PQT for the following comparison between agents.

The results of the comparison between agents on the "Learn to count" task are shown in Figure 3. We find that PQT provides a stronger training signal, allowing the Neural agent to outperform the Evolutionary agent given enough training. However, the Evolutionary agent is learning faster during the initial 1000 samples, showing higher sample efficiency. Evo-NAS agent initial quick improvement shows its ability to take advantage of the sample efficiency of Evolution. We also observe that learning the proper mutation patterns allows the Evo-NAS agent to find good sequences faster than the Evolutionary agent. Hence, in sample efficiency the Evo-NAS agent strongly outperforms both the Evolutionary and the Neural agent.

Figure 4: Results of the experiments on 7 text classification tasks. Evo-NAS and Neural agents use PQT with learning rate 0.0001 and entropy penalty 0.1. Their controllers are trained to maximize log likelihood of the top 5 trials. Each experiment was ran 10 times. Each run was bound to 2h of runtime in which 30 trials were trained in parallel. For each run, we have selected the model that obtained the best ROC-AUC on the validation set (the best reward). These best models were then evaluated by computing the ROC-AUC on the holdout testset. The empty circles in the plot represent the test ROC-AUC achieved by each of the 10 best models. The filled circles represent the means of the empty circles. We superpose $\pm 1$ standard deviation bars.

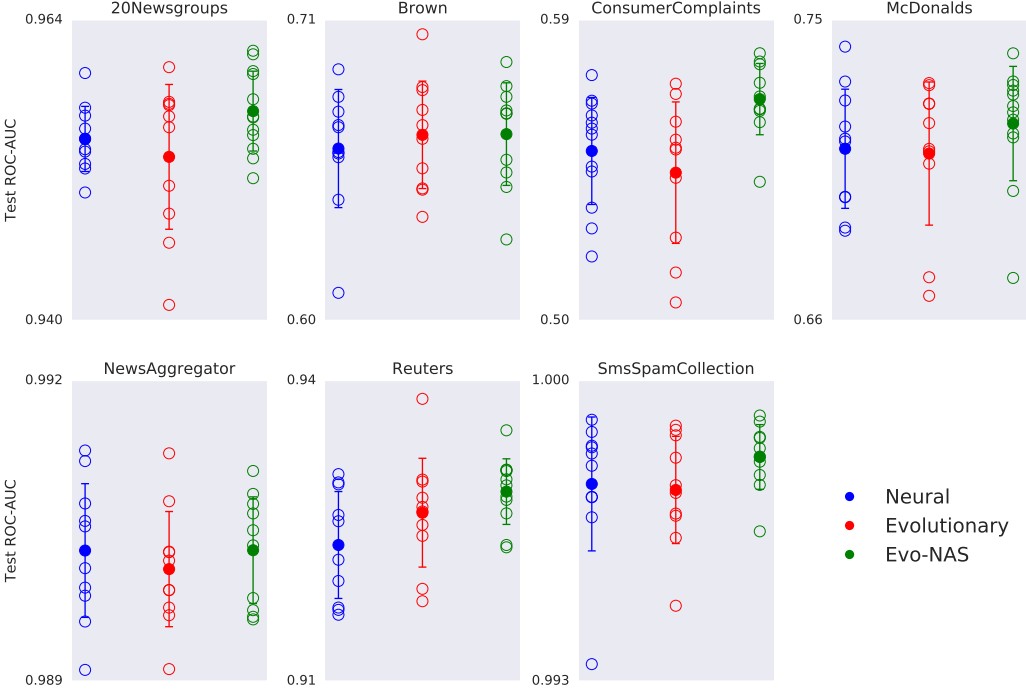

## 5.2 TEXT CLASSIFICATION TASKS

We evaluate the Neural, Evolutionary and Evo-NAS agents on the task of finding architectures for 7 public text classification tasks.

Similarly to (Wong et al., 2018), we design a search space of common architectural and hyperparameter choices that define two-tower feedforward neural networks (FFNN) child models. These architectures are similar to the "wide and deep" models (Cheng et al., 2016). One tower is a deep FFNN, built by stacking: a pre-trained text-embedding module, a stack of fully connected layers, and a softmax classification layer. The deep tower is regularized with L2 loss. The other tower is a wide-shallow layer that directly connects the one-hot token encodings to the softmax classification layer with a linear projection. The wide tower is regularized with L1 loss to promote sparsity. The wide tower allows the model to learn task-specific biases for each token directly, such as trigger words, while the deep tower allows it to learn complex patterns.

The controller defines the child model architecture by selecting a value for every available architectural or hyper-parameter choice. The details of the search space are reported in Table 3. The search space defines the sequence of actions and the values available for each action. The first action selects the pre-trained text-embedding module. The details of the text-embedding modules are reported in Table 4. These modules are available via the TensorFlow Hub service[1]. Using pretrained text-embedding modules has two benefits: first, improves the quality of the child models trained on smaller datasets, and second, decreases convergence time of the child models. Another particularity

---

[1]https://www.tensorflow.org/hub

Table 2: Best ROC-AUC(%) on the testset for each algorithm and dataset. We report the average over 10 runs, as well as $\pm$ 2 standard-error-of-the-mean (s.e.m.) Bolding indicates the best performing algorithm or those within 2 s.e.m. of the best.

| Dataset (reference) | Neural | Evolutionary | Evo-NAS |
|---|---|---|---|
| 20Newsgroups (Lang, 1995) | $95.45 \pm 0.17$ | $95.31 \pm 0.39$ | $\mathbf{95.67 \pm 0.19}$ |
| Brown (Francis & Kuera, 1982) | $\mathbf{66.29 \pm 1.44}$ | $\mathbf{66.79 \pm 1.31}$ | $\mathbf{66.82 \pm 1.25}$ |
| ConsumerComplaints (catalog.data.gov) | $55.08 \pm 0.97$ | $54.43 \pm 1.41$ | $\mathbf{56.63 \pm 0.71}$ |
| McDonalds (crowdflower.com) | $\mathbf{71.14 \pm 1.19}$ | $\mathbf{71.00 \pm 1.43}$ | $\mathbf{71.90 \pm 1.03}$ |
| NewsAggregator (Lichman, 2013) | $\mathbf{99.03 \pm 0.04}$ | $\mathbf{99.01 \pm 0.04}$ | $\mathbf{99.03 \pm 0.04}$ |
| Reuters (Debole & Sebastiani, 2004) | $92.36 \pm 0.36$ | $\mathbf{92.68 \pm 0.36}$ | $\mathbf{92.89 \pm 0.21}$ |
| SmsSpamCollection (Almeida et al., 2011) | $99.76 \pm 0.10$ | $99.75 \pm 0.08$ | $\mathbf{99.82 \pm 0.05}$ |

is that the optimizer for the deep column can be either Adagrad (Duchi et al., 2011) or Lazy Adam [2]. "Lazy Adam" refers to a commonly used version of Adam (Kingma & Ba, 2014) that computes the moving averages only on the current batch. These are efficient optimizers, that allow to halve the back-propagation time, compared to more expensive optimizers such as Adam. The optimizer used for the wide column is the standard FTRL (McMahan, 2011). Notice that this search space is not designed to discover original architectures that set a new state-of-the-art for this type of tasks, but it is rather a medium complexity architecture search environment, that allows to analyze the properties of the agents.

All the experiments in this section are run with a fixed budget: 30 trials are trained in parallel for 2 hours with 2 CPUs. Choosing a small budget allows to run a higher number of replicas for each experiment, to increase the significance of the results. It also makes the budget accessible to most of the scientific community, thus increasing the reproducibility of the results.

During the experiments, the models sampled by the agent are trained on the training set of current text classification task, and the Area under the ROC curve (ROC-AUC) on the validation set is used as the reward for the agent. As the final unbiased evaluation metric, we measure the ROC-AUC computed on the holdout testset, for the child model that achieved the best reward. For the datasets that do not come with a pre-defined train/validation/test split, we split randomly 80/10/10.

We use ROC-AUC as the main metric, since it provides a less noisy reward signal, compared to the more commonly used accuracy. In a preliminary experiment, we validated this hypothesis by running experiments on the ConsumerComplaints task. Then, for a sample of 30 models, we have computed 4 metrics: ROC-AUC on validation and test set, accuracy on validation and test set. The Pearson correlation between the validation ROC-AUC and the test ROC-AUC resulted to be 99.96%, while between the validation accuracy and the test accuracy resulted to be 99.70%. The scatter plot of these two sets is reported in Figure 6.

We validate the results of the comparison between PQT and Reinforce done in Section 5.1 by running 5 experiment replicas for each of the 7 tasks using the Neural controller with both training algorithms. We measure an average relative gain of $+1.13\%$ over the final test ROC-AUC achieved by using PQT instead of Reinforce. Thus, we assume that the gains observed by using PQT instead of Reinforce generalize.

We will use PQT for all the following experiments to train the Evo-NAS and Neural agents to maximize the log likelihood of the top 5 trials. For the Evo-NAS and Evolutionary agents, we have set the exploitation probability $p$ to 0.5.

To measure the quality of the models generated by the three agents, we run 10 experiment replicas for each of the 7 tasks, and we measure the test ROC-AUC obtained by the best model generated by each experiment replica. The results are summarized in Table 2 and Figure 4. The Evo-NAS agent

---

[2]https://www.tensorflow.org/api_docs/python/tf/contrib/opt/LazyAdamOptimizer

Figure 5: Reward moving average for different agents. The average is computed over a window of 50 consecutive trials. We ran 3 replicas for each experiment. The shaded area represents minimum and maximum value of the rolling average across the runs.

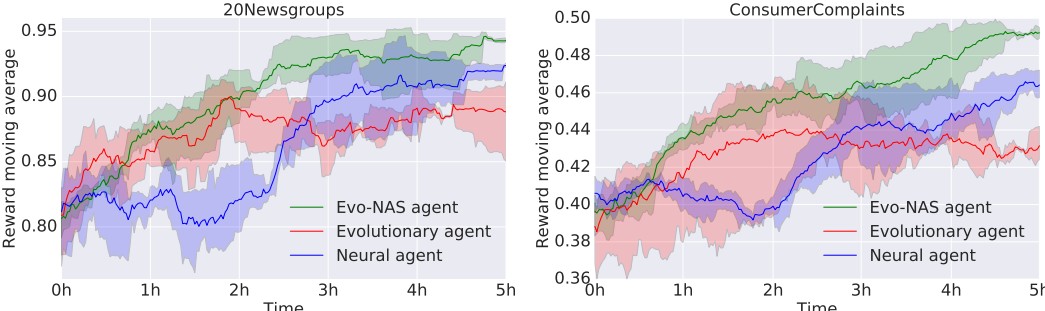

achieves the best performance on all 7 tasks. On 3 out of 7 tasks it significantly outperforms both the Neural and the Evolutionary agents.

We also report the number of trials each of the agents performed during the 2h long experiments, and summarize the results in Table 5 and Figure 7. We find that the Evolutionary and the Evo-NAS agents strongly outperform the Neural agent in terms of number of trials performed. The Evo-NAS agent achieves the largest number of trials on 6 out of 7 datasets, while the Evolutionary agent on 5 out of 7 datasets. On 4 datasets the Evolutionary and Evo-NAS agents perform joint best. We conclude that this shows that the evolutionary algorithms are biased towards faster models, as shown in (Real et al., 2018).

An in depth analysis of the architectures that achieved the best performance is beyond the scope of this paper. However, we want to mention a few relevant patterns, that emerge across tasks. The agent often selects FFNNs with fewer, but wider, layers for the deep part of the network. The learning rate for the best models for both wide and deep parts is in the bottom of the range (0.001). The controller also prefers to disable L1 and L2 regularization. Our interpretation is that reducing the number of parameters is a simpler and more effective regularization, that is preferred over adding L1 and L2 factor to the loss.

To verify that the learning patterns highlighted in Section 5.1 generalize, we plot in Figure 5 the reward moving average for two selected tasks: 20Newsgroups and ConsumerComplaints. For these experiments, we have extended the time budget from 2h to 5h. This time budget extension is needed to be able to capture longer term trends exhibited by the Neural and Evo-NAS agents. We run 3 replicas for each task. In the early stages of the experiments, we notice that the quality of the samples generated by the Neural agent are on the same level as the randomly generated samples, while the quality of the samples generated by the Evo-NAS and Evolutionary agents grows steadily.

In the second half of the analyzed experiments, we notice that the Neural agent starts applying the learnt patterns to the generated samples. The quality of the samples generated by the Evolutionary agent flattens, which we assume is due to the fact that the quality of the samples in the population is close to optimum, and the quality of the samples cannot improve, since good mutations patterns cannot be learned. Finally, we observe that the Evo-NAS agent keeps generating better, samples or even improves.

## 6 CONCLUSION

We introduce a class of Evo-NAS hybrid agents, which are designed to retain both the sample efficiency of evolutionary approaches, and the ability to learn complex patterns of Neural agents. We experiment on synthetic and text classification tasks, analyze the properties of the proposed Evo-NAS agent, and show that it can outperform both Neural and Evolutionary agents. Additionally, we show that Priority Queue Training outperforms Reinforce also on architecture search applications.

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

## A    APPENDIX

Table 3: The search space defined for text classification experiments.

| Parameters | Search space |
|---|---|
| 1) Input embedding modules | See Table 4 |
| 2) Fine-tune input embedding module | {True, False} |
| 3) Use convolution | {True, False} |
| 4) Convolution embedding dropout rate | {0.0, 0.1, 0.2, 0.3, 0.4} |
| 5) Convolution activation | {relu, relu6, leaky relu, swish, sigmoid, tanh} |
| 6) Convolution batch norm | {True, False} |
| 7) Convolution max ngram length | {2, 3} |
| 8) Convolution dropout rate | {0.0, 0.1, 0.2, 0.3, 0.4} |
| 9) Convolution number of filters | {32, 64, 128} |
| 10) Number of hidden layers | {0, 1, 2, 3, 5} |
| 11) Hidden layers size | {64, 128, 256} |
| 12) Hidden layers activation | {relu, relu6, leaky relu, swish, sigmoid, tanh} |
| 13) Hidden layers normalization | {none, batch norm, layer norm} |
| 14) Hidden layers dropout rate | {0.0, 0.05, 0.1, 0.2, 0.3, 0.4, 0.5} |
| 15) Deep optimizer name | {adagrad, lazy adam} |
| 16) Lazy adam batch size | {128, 256} |
| 17) Deep tower learning rate | {0.001, 0.005, 0.01, 0.05, 0.1, 0.5} |
| 18) Deep tower regularization weight | {0.0, 0.0001, 0.001, 0.01} |
| 19) Wide tower learning rate | {0.001, 0.005, 0.01, 0.05, 0.1, 0.5} |
| 20) Wide tower regularization weight | {0.0, 0.0001, 0.001, 0.01} |
| 21) Number of training samples | {100000, 200000, 500000, 1000000, 2000000, 5000000} |

Table 4: Options for text input embedding modules. These are pre-trained text embedding tables, trained on datasets with different languages and size. The text input to these modules is tokenized according to the module dictionary and normalized by lower-casing and stripping rare characters. We provide the handles for the modules that are publicly distributed via the TensorFlow Hub service (https://www.tensorflow.org/hub).

| Language/ID | Dataset size (tokens) | Embed dim. | Vocab. size | Training algorithm | TensorFlow Hub Handles Prefix: https://tfhub.dev/google/ |
|---|---|---|---|---|---|
| English-small | 7B | 50 | 982k | Lang. model | nnlm-en-dim50-with-normalization/1 |
| English-big | 200B | 128 | 999k | Lang. model | nnlm-en-dim128-with-normalization/1 |
| English-wiki-small | 4B | 250 | 1M | Skipgram | Wiki-words-250-with-normalization/1 |
| Universal-sentence-encoder | - | 512 | - | (Cer et al., 2018) | universal-sentence-encoder/2 |

Figure 6: Correlation of validation accuracy with test accuracy (Left) and validation ROC-AUC with test ROC-AUC (Right). The correlation is higher for ROC-AUC. For plotting the correlations, we used the ConsumerComplaints dataset.

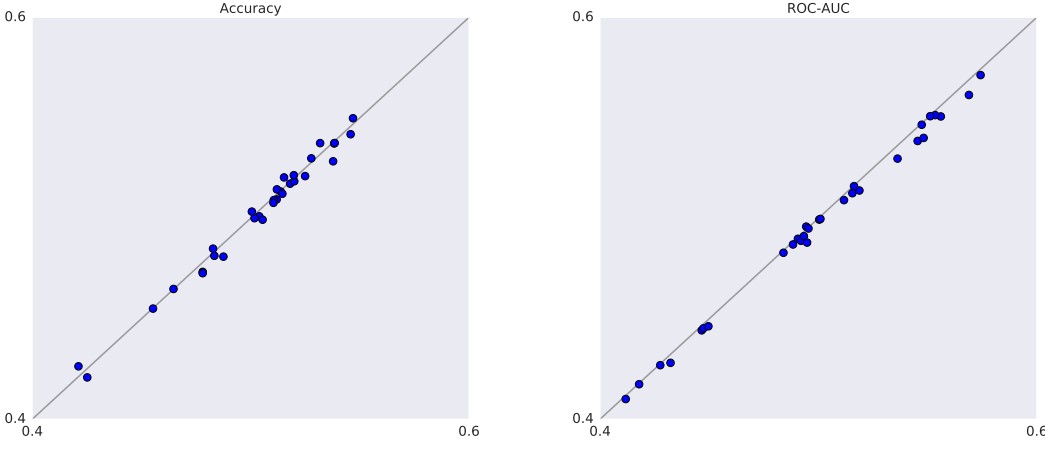

Figure 7: Number of trials performed for the experiments from Figure 4. The empty circles represent the number of trials perfomed in each of the 10 experiments. The filled circles represent the means of the empty circles. We superpose $\pm 1$ standard deviation bars.

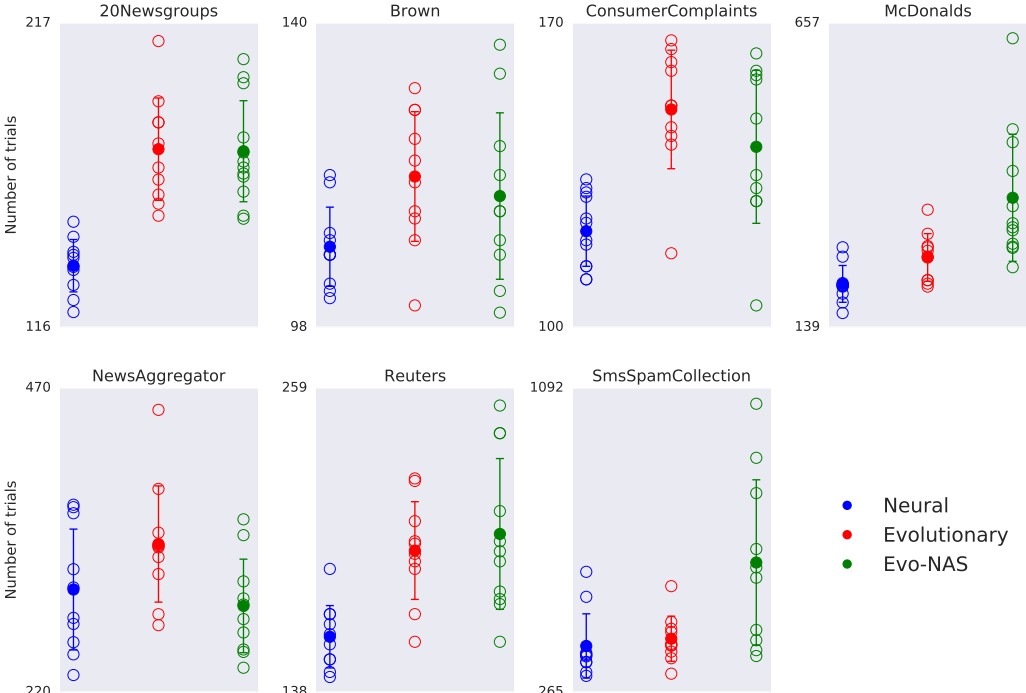

Table 5: The number of trials performed for the experiments from Figure 4. We report the average over 10 runs, as well as $\pm$ 2 standard-error-of-the-mean (s.e.m.). Bolding indicates the algorithm with the highest number of trials or those that have performed within 2 s.e.m. of the largest number of trials.

| Dataset (reference) | Neural | Evolutionary | Evo-NAS |
|---|---|---|---|
| 20Newsgroups (Lang, 1995) | $136 \pm 6$ | $\mathbf{175 \pm 11}$ | $\mathbf{174 \pm 10}$ |
| Brown (Francis & Kuera, 1982) | $109 \pm 4$ | $\mathbf{119 \pm 6}$ | $\mathbf{116 \pm 8}$ |
| ConsumerComplaints (catalog.data.gov) | $122 \pm 5$ | $\mathbf{150 \pm 9}$ | $142 \pm 12$ |
| McDonalds (crowdflower.com) | $213 \pm 21$ | $258 \pm 27$ | $\mathbf{359 \pm 65}$ |
| NewsAggregator (Lichman, 2013) | $304 \pm 33$ | $\mathbf{342 \pm 32}$ | $291 \pm 26$ |
| Reuters (Debole & Sebastiani, 2004) | $160 \pm 8$ | $\mathbf{194 \pm 13}$ | $\mathbf{201 \pm 19}$ |
| SmsSpamCollection (Almeida et al., 2011) | $390 \pm 58$ | $410 \pm 41$ | $\mathbf{617 \pm 150}$ |

Table 6: Statistics of the text classification tasks.

| Dataset | Train samples | Valid samples | Test samples | Classes | Lang | Mean text length |
|---|---|---|---|---|---|---|
| 20 Newsgroups | 15,076 | 1,885 | 1,885 | 20 | En | 2,000 |
| Brown Corpus | 400 | 50 | 50 | 15 | En | 20,000 |
| Consumer Complaints | 146,667 | 18,333 | 18,334 | 157 | En | 1,000 |
| McDonalds | 1,176 | 147 | 148 | 9 | En | 516 |
| News Aggregator | 338,349 | 42,294 | 42,294 | 4 | En | 57 |
| Reuters | 8,630 | 1,079 | 1,079 | 90 | En | 820 |
| SMS Spam Collection | 4,459 | 557 | 557 | 2 | En | 81 |

