# OpenReview forum: "Evolutionary-Neural Hybrid Agents for Architecture Search"
_ICLR.cc/2019/Conference_

### Official Review · AnonReviewer3 · 2018-11-01
**Intuitive idea, unclear explanation.**

**Rating:** 4
**Confidence:** 4

**Review:**

The paper proposes a class of Evolutionary-Neural hybrid agents (Evo-NAS) to take advantage of both evolutionary algorithms and reinforcement learning algorithms for efficient neural architecture search.

1. Doesn't explain how exactly the mutation action is learned, and missing the explanation of how RL acts on its modification on NAS (Evo-NAS).
2. Very poor explanation on LEARN TO COUNT experiment. The experiment contains difficult setups on a toy data, which makes it difficult to repeat. In figure 3, the paper says that the sample efficiency of the Evo-NAS strongly outperforms both the evolutionary and the neural agent. However, where the strength comes from is not discussed in detail. In figure 2, the paper claims that PQT outperforms Reinforce for both the Neural and the Evo-NAS agent. For the Evo-NAS agent, the gain is especially pronounced at the beginning of the experiment. Thus, the paper concludes that PQT can provide a stronger training signal than Reinforce. However, how much stronger training signal can obtain of the proposed method is not discussed. Because the experiments of 5.1 is setup on a toy data with complicated parameters. The conclusions based on this data set is not convincing. It would be better to add comparative results on the CIFAR and Imagenet data for convenient comparisons with state-of-the-art.
3. Confusing notation and experimental setup. In 5.1, the sequence a is first defined as <a1, a2, .., an>. Then, after eq.2, the sequence a is given as a=<1, 2, ..., n>. It would be better to use different symbols here.

---

> ### Author Response · Authors · 2018-11-10
> **Response to AnonReviewer3**
>
> Thank you so much for your feedback!
>
> We tried to write everything as clearly as possible, could you please tell us exactly which things were unclear? We would be especially interested in hearing what you meant in points 2 and 3 in your review.

---

### Official Review · AnonReviewer2 · 2018-11-02
**Interesting method. However, empirical results are not convincing enough.**

**Rating:** 4
**Confidence:** 4

**Review:**

Summary:

The paper proposes a hybrid approach which combines evolution and RL. The key idea is to conduct tournament selection over a population of architectures with learned mutations. The mutations are defined as the output of an RNN controller which either reuses or alters the sequence descriptor of the parent at each step. The proposed hybrid architect is evaluated on both synthetic and text classification tasks and then compared against pure evolutionary and RL-based agents.

Pros:

* The method can be viewed as a generalization of conventional evolution by replacing the handcrafted (uniform) distribution of mutations with a learned one. On the one hand, this should hopefully improve the sample efficiency of pure genetic methods since the population can evolve towards more meaningful directions, assuming useful patterns can be learned by the mutation controller. On the other hand, mutating existing architectures seem a easier task than sampling the entire architecture from scratch.
* The synthetic experiment is interesting, though it's hard to draw any conclusions based a single task.

Cons:

* To my knowledge, all text classification tasks used in 5.2 are quite small. There is no evidence that the method can scale to and work well on large-scale tasks, where improving the sample efficiency becomes truly crucial and challenging.
* It is good to see comparisons against pure evo and RL within the authors' own search space. However, the advantage of the proposed evo-NAS, especially when evaluated on real-world text classification tasks, does not seem significant enough. In particular, there is a clear overlap between the performance of architectures found by NAS, evo and evo-NAS (Figure 4). The advantage of evo-NAS is even smaller if we compare the very best model (as can be read from Figure 4) instead of the average among the top 10 (as reported in Table 2). In my option, performance of the strongest model is arguably more interesting than the averaged one in practice.
* Since no results on CIFAR or ImageNet are provided as in most prior works in the literature, it is impossible to empirically compare the method with the state-of-the-art. The experiments would be more convincing if a comparison can be provided on those benchmarks. Otherwise, it is possible that the current search space & hyperparameters are tailored towards evo-NAS and it remains unclear whether the method can generalize well to other domains and/or search spaces.

---

### Official Review · AnonReviewer1 · 2018-11-07
**Simple and intuitive idea, insufficiently convincing results**

**Rating:** 5
**Confidence:** 2

**Review:**

Review:
The paper introduces a novel way to do architecture search that uses an RNN to guide the mutation operation. The method and the motivation of the idea as long with the related work are all clearly described. However, the experiments section does not show a big uplift of the method versus the baselines and the number of types of tasks is relatively small (artificial and text).

Cons:
- No image task
- No large scale task to show the scalability
- No baselines that are not coming from AUTO-ML to show the relative performance of a classical method

---

### Meta-Review · Area_Chair1 · 2018-12-14

**Confidence:** 4
**Recommendation:** Reject

**Metareview:**

Reviewers are in a consensus and recommended to reject after engaging with the authors. Please take reviewers' comments into consideration to improve your submission should you decide to resubmit.